# Ti–15Zr and Ti–15Zr–5Mo Biomaterials Alloys: An Analysis of Corrosion and Tribocorrosion Behavior in Phosphate-Buffered Saline Solution

**DOI:** 10.3390/ma16051826

**Published:** 2023-02-23

**Authors:** Adriana Alencar Santos, Jean Valdir Uchôa Teixeira, Carlos Alberto Fonzar Pintão, Diego Rafael Nespeque Correa, Carlos Roberto Grandini, Paulo Noronha Lisboa-Filho

**Affiliations:** 1Materials Science and Technology Program, Department of Physics, School of Sciences, UNESP—São Paulo State University, Bauru 17033-360, SP, Brazil; 2Physical and Rheological Characterization Laboratory, Department of Physics, School of Sciences, UNESP—São Paulo State University, Bauru 17033-360, SP, Brazil; 3Laboratory of Anelasticity and Biomaterials, Department of Physics, School of Sciences, UNESP—São Paulo State University, Bauru 17033-360, SP, Brazil; 4Laboratory of Nanotechnology and Advanced Materials Department of Physics, School of Sciences, UNESP—São Paulo State University, Bauru 17033-360, SP, Brazil

**Keywords:** biomaterial titanium alloys, corrosion, EIS, tribocorrosion

## Abstract

It is crucial for clinical needs to develop novel titanium alloys feasible for long-term use as orthopedic and dental prostheses to prevent adverse implications and further expensive procedures. The primary purpose of this research was to investigate the corrosion and tribocorrosion behavior in the phosphate buffered saline (PBS) of two recently developed titanium alloys, Ti–15Zr and Ti–15Zr–5Mo (wt.%) and compare them with the commercially pure titanium grade 4 (CP–Ti G4). Density, XRF, XRD, OM, SEM, and Vickers microhardness analyses were conducted to give details about the phase composition and the mechanical properties. Additionally, electrochemical impedance spectroscopy was used to supplement the corrosion studies, while confocal microscopy and SEM imaging of the wear track were used to evaluate the tribocorrosion mechanisms. As a result, the Ti–15Zr (α + α′ phase) and Ti–15Zr–5Mo (α″ + β phase) samples exhibited advantageous properties compared to CP–Ti G4 in the electrochemical and tribocorrosion tests. Moreover, a better recovery capacity of the passive oxide layer was observed in the studied alloys. These results open new horizons for biomedical applications of Ti–Zr–Mo alloys, such as dental and orthopedical prostheses.

## 1. Introduction

Due to its high strength, low Young’s modulus, superior corrosion resistance, and biocompatibility, titanium and its alloys are widely employed in various biomedical applications [1]. However, implants have limitations and consequent side effects that seriously impact health during long-term usage [2]. The metallic implants have various drawbacks, usually due to low wear resistance under chewing and contact with corrosive physiological fluids [3]. The phosphate-buffered saline solution (PBS) is a reference inorganic solution composition that simulates the corrosiveness of body fluids in order to evaluate metallic materials as prospective biomaterials [4,5]

The α- and β-phases are the two polymorphic phases in Ti alloys. By adding alloying elements and using thermomechanical processes, the hexagonal close-packed (hcp; α-phase) is transformed into a body-centered cubic (bcc; β-phase) structure [6]. Studies have been performed on novel biomedical Ti alloys in response to potential downsides of ion releasing, especially adding non-toxic and non-allergic alloying elements like Mo, Nb, Ta, and Zr [7]. Zr is a neutral stabilizer that has the benefit of hardening the alloy, which is desirable for long-term implantation to prevent the release of debris by mechanical friction [8]. In addition, adding Zr in Ti alloys improves the mechanical, corrosion, and biocompatibility properties [9]. Contrarily, Mo is an isomorphous β-stabilizer element in Ti alloys, primarily used to manufacture non-toxic and non-allergic β-type Ti alloys for implants [10]. Ti–Mo alloys depict superior corrosion, mechanical strength, and minor Young’s modulus [11,12]. However, as Mo is a refractory metal, its use as an alloying element is limited by the melting point. Ti–Zr–Mo alloys have attracted attention in this context, since Zr can assist the β phase precipitation when added to Mo [13].

Furthermore, it is crucial to conduct studies on the corrosion and tribocorrosion behaviors of newly developed materials that might substitute commercial ones. Through tribocorrosion and corrosion processes, it is possible to observe that corrosion is accelerated by wear particles in implants due to the interaction with the human body [14]. Additionally, when there is an oxide layer and debris from tribocorrosion chemical reactions in the body, patients may experience periprosthetic bone loss. As a result, bone attachment loss and inflammatory responses, adverse tissue reactions, metal poisoning, and carcinogenesis occur. Consequently, these issues highlight the increased importance of tribocorrosion research [15].

The mechanism of TiO_2_ formation in Ti-based alloys interacting with the fluid in human bodies, which is constantly present in oxidizing conditions, is responsible for outstanding biocompatibility [16]. This naturally formed oxide layer protects titanium implants from corrosion and decreases toxic ionic from the alloy released into surrounding living tissues [17].

On tribocorrosion tests, metal-on-metal (e.g., stainless steel on Ti surfaces) produced more abrasion and adhesion wear than ceramic-on-metal (e.g., alumina on Ti surfaces), which raised the wear rate. Additionally, in ceramic-on-metal wear experiments, when comparing loads, the wear rate was higher at a load of 5 N and a stroke length of 5 mm than it was at 2 N and 10 mm, respectively; this provides reasonable evidence for abrasion wear mechanism and material transfer, since more wear debris was accumulated [18].

While the mechanical properties of Ti–Zr alloys depend on their microstructure and phase composition, their chemical or electrochemical stability is primarily influenced by the proportions of zirconium and titanium. Therefore, controlling the mechanical, chemical, electrochemical, and biological properties of Ti–Zr alloys necessitates an in-depth knowledge of the phase, manufacturing process, and surface treatment [19].

Therefore, this research examined the corrosion and tribocorrosion behavior of Ti in a simulated physiological environment. Two alloy groups were investigated and compared to the reference material. Two alloys were produced using arc melting (Ti–15Zr and Ti–15Zr–5Mo; wt.%), while a commercially pure Ti grade 4 (CP–Ti G4) served as a control group. Corrosion and tribocorrosion experiments were carried out on those samples in PBS.

## 2. Materials and Methods

### 2.1. Materials

Before the development of both Ti–15Zr or Ti–Zr (binary) and Ti–15Zr–5Mo or Ti–Zr–Mo (ternary), the raw material of commercially pure Ti grade 2 cylindrical bars (Aldrich Inc., São Paulo, Brazil, 99.7% purity), pure Zr foil (Aldrich Inc., São Paulo, Brazil, 99.8% purity), and pure Mo wire (Aldrich Inc., São Paulo, Brazil, 99.9%) were weighed and separated to establish the proportionality of each alloy’s composition. The commercially pure Ti grade 4 (CP–Ti G4) bar (Acnis do Brasil., Sorocaba, Brazil, 98.56% purity) was acquired for comparison.

### 2.2. Sample Processing

The Ti–Zr and Ti–Zr–Mo samples were produced in an arc-melting furnace under an argon (Ar) environment. Ingots were formed by melting the raw materials in a water-cooled copper crucible, using a tungsten electrode. Afterward, the ingots were subjected to two heat treatments in a tubular resistive furnace under a 10^−5^ Torr vacuum. Initially, they were homogenized at 1000 °C for 24 h, with a heating rate of 10 °C·min^−1^ and cooled to room temperature at a rate of 10 °C·min^−1^. Then, they were swaged using a rotary press SWAGE, 3F 1000 (FENN, East Berlin, CT, USA), at 1000 °C with air cooling to form rods (ϕ = 11 mm). Finally, the rods were submitted to solution treatment, in which they were heated up to 400 °C at a rate of 10 °C·min^−1^. Then, the temperature was kept constant for 4 h to eradicate microstructural defects. Finally, they were cooled to room temperature at a rate of 10 °C·min^−1^.

### 2.3. Sample Preparation

The samples were cut into 3 mm thick discs using an cutter (Actspark, model SP1, São Paulo, Brazil) by electro-erosion with molybdenum wire. The samples’ surfaces undergo metallographic techniques, including bakelite inlay and sanding using waterproof silicon carbide (SiC) sandpapers, up to 1200 mesh. Before the corrosion and tribocorrosion tests, the samples were subjected to an ultrasonic bath with Milli-Q water (Merk group, Cotia, Brazil) for 30 min.

### 2.4. Density Measurements

An Oanalytical balance (Ohaus Corporation, model Explorer, Parsippany-Troy Hills, NJ, USA) with a density kit was used in this experiment. The density values were calculated from Archimedes’ principle and theoretical ones by the weighted average densities of the alloying elements.

### 2.5. Chemical Analysis

Energy dispersive X-ray fluorescence spectroscopy (XRF) was utilized to conduct the chemical analysis using EDX-720 equipment (Shimadzu, Barueri, Brazil). To identify the type and quantity of the elements in the sample, XRF irradiates the sample with X-rays measuring the energy (wavelength) and intensity of the fluorescent X-rays emitted.

### 2.6. Phase Composition

X-ray diffraction (XRD) was used to determine the crystallographic phases using a D/Max diffractometer (Rigaku, Tokyo, Japan) with a Cu–Kα radiation source (λ = 1.54 Å), running at 40 kV and 15 mA. The divergence slit for the continuous scanning mode was configured at 1/4° in a θ–2θ configuration. The measurements were taken in the fixed time mode at a grazing angle (ω) of 3°, a step size of 0.02°, and a duration per step of 3 s.

### 2.7. Microstructural Characterizations

Optical microscopy (OM; BX51M microscope, Olympus, São Paulo, Brazil) and scanning electron microscopy (SEM; model EVO-LS15, Carl Zeiss, Jena, Germany) were used to reveal the microstructural characteristics. The samples were first placed through the usual metallographic processes, including SiC waterproof sandpaper grinding until #2000, polishing with colloidal alumina suspensions (0.25 m), and etching in Kroll’s solution.

### 2.8. Microhardness Tests

Vickers microhardness was measured using a durometer (Struers LLC., model Duramin-40 AC3, Cleveland, OH, USA) with a load of 10 gf and dwell time of 10 s. An amount of 2 indentation patterns were collected with a total of 13 points, including the first pattern with 7 points along the horizontal direction and the second with 6 points in the vertical direction.

### 2.9. Electrochemical Tests

An electrochemical cell was assembled, wherein the samples were embedded in a three-electrode configuration to perform corrosion tests. The CP–Ti G4 sample with a surface area of 1.26 cm^2^ and the Ti–Zr and Ti–Zr–Mo samples with 0.95 cm^2^ were set up as the working electrodes, a platinum wire was the counter electrode, and an Ag/AgCl electrode was the reference. The electrochemical cell was immersed in 100 mL of PBS (pH = 7.4, Sigma-Aldrich Merck (São Paulo, Brazil), at a physiological temperature of 37 °C, controlled by a heater. Autolab potentiostats (Metrohm, São Paulo, Brazil,) and the NOVA 2.3 software were used for electrochemical measurements. Prior research conducted by Trino et al. [20] was used to develop the electrochemical methodology. Initially, the open-circuit potential (OCP) was measured to stabilize the sample surface under the electrochemical cell’s experimental conditions. Subsequently, linear polarization was performed to investigate the effect of natural corrosion. The corrosion potential (E_corr_), corrosion current density (I_corr_), polarization resistance (R_p_), and corrosion rate (CR) were determined using Tafel’s extrapolation method, ranging from −1 V to 2 V, with a scan rate of 10 mV·s^−1^. All the electrochemical tests were performed in duplicate.

### 2.10. EIS Measurement

Electrochemical impedance spectroscopy (EIS) was applied to evaluate the passive film integrity and surface stability upon electrolyte interaction. EIS was recorded at frequencies ranging from 106 to 10–1 Hz, using a sinusoidal voltage amplitude of 10 mV and frequencies of 10 Hz for points per decade. The EIS data were analyzed with the Metrohm Autolab potentiostat’s NOVA program. The Nyquist and Bode plots were constructed employing the real (Z′), imaginary (Z″), total (|Z|), and phase angle. In addition, parameters such as solution resistance (Rs), polarization resistance (RP), constant phase elements (CPE), and charge transfer coefficient (α) were calculated to determine the kinetics of the electrochemical reactions between the metallic surface and the PBS solution.

The RP values in corrosion tests were obtained by the NOVA program using the Butler–Volmer equation with the results of Tafel’s extrapolation (Equation (1)), where *I* is current density; *η* is polarization; and *βa* and *βc* are anodic and cathodic Tafel’s slopes, respectively. Equation (2) was used to calculate RP [21].
(1)$$I=Icorr\;\left(e^{\left(\frac{\eta }{\beta a}\right)}-e^{\left(\frac{-\eta }{\beta c}\right)}\right)$$
(2)$$\frac{1}{Rp}=Icorr\;\left(\frac{1}{\beta a}+\frac{1}{\beta c}\right)$$

The corrosion rates of polarization, *CR* (mm·yr^−1^), were determined using Equation (3), where *M*, *F*, *S*, *d*, and *t* are the equivalent molar weight (g·mol^−1^), Faraday constant (96,500 C·mol^−1^), the surface area of the electrode, density (g·cm^−3^), and immersion time, respectively [22].
(3)$$CR=\frac{Icorr\cdot t\cdot \;M}{F\cdot S\cdot d}$$

### 2.11. Tribocorrosion Tests

Tribocorrosion tests were performed to investigate the effects of friction processes in a biological environment using PBS. The samples were placed at the bottom of an electrochemical cell in a two electrodes configuration. The sample and an Ag/AgCl electrode were used as a working electrode and reference, respectively. An amount of 50 mL of PBS was added to the electrochemical cell and heated to 37 °C. Tribocorrosion experiments were carried out using a ball-on-rotating plate tribometer. The sample was maintained vertically against an alumina ball (Al_2_O_3_) (Anton Paar Tritec SA, São Paulo, Brazil), which has a composition of 99.80%, a diameter of 6 mm, a hardness of 1.650, and a roughness of 0.02 m with a counter body that had a load of 1.5 N. The chosen load was intended to imitate chewing, since the biomaterials investigated are applied in dental implants [23].

In each sample, three distinct potential zones were shown. First, the stabilization zone corresponded to the period before sliding to the natural equilibrium between the surface and solution. The samples of each alloy were kept in PBS until the OCP was stabilized. In the second zone, the sliding friction potential corresponded to the period of mechanical wear when fretting started. The sliding friction was maintained for 30 min, and the coefficient of friction (COF) between the alumina sphere and the samples’ surface was determined. The passivation was maintained until the OCP stabilized for approximately 30 min. All the tribocorrosion tests were performed in duplicate.

In addition, the two-dimensional (2D) topography of the wear tracks after tribocorrosion testing was examined using a confocal microscope DCM3D (LEICA, Wetzlar, Germany). The volume loss through the profile was calculated using Equation (4), where *Vm*, Rm, and Am are the average volume loss (mm^3^), average radius (2 mm), and average area (mm^2^), respectively [24].
(4)Vm=2∗π·Rm∗Am

The wear rate was determined using Equation (5), where *Wr*, *Vm*, and *E_T_* are wear rate (mm^3^·N^−1^·m^−1^), an average of volume loss (mm^3^), and total energy (N·m), respectively. The wear rate is the *V* divided by *E_T_*, as shown [25]:(5)$$W_{r}=\frac{Vm}{E_{T}}$$

## 3. Results and Discussion

### 3.1. Density Measurements

Figure 1 shows that the density values of CP–Ti G4, Ti–Zr, and Ti–Zr–Mo samples agreed with the theoretical values, as they were close to 4.51 g·cm^−3^, 4.72 g·cm^−3^ and 4.86 g·cm^−3^, respectively [26]. Since the theoretical densities were acquired from the weighted average of the individual densities of each element to its nominal composition, it can be inferred that the produced binary and ternary samples had good quality for the study.

### 3.2. Chemical Analysis

The chemical compositions of the developed alloys and the titanium acquired (CP–Ti G4) were validated using XRF and are shown in Table 1. As a result, the content of both binary and ternary alloys were in excellent agreement with the nominal composition; While the ternary Mo is slightly higher, it is within the fluctuation and resolution of the techniques. Some impurities, such as Fe, were detected in CP–Ti G4, which is consistent with the technical datasheet of commercially pure titanium grade 4 (Acnis do Brasil., Sorocaba, Brazil, 98.56% purity), according to ASTM F67 [27].

### 3.3. Phase Composition

Figure 2 shows the XRD patterns of the Ti–Zr, Ti–Zr–Mo, and CP–Ti G4 samples. The CP–Ti G4 sample exhibited peaks corresponding to a single α-Ti phase (JCPDS 00-044-1294 card). This is expected, considering it is a commercially pure grade of Ti with a stable hexagonal structure at room temperature in its metallic form.

Similarly, the Ti–Zr exhibited the same pattern as CP–Ti G4. However, Correa et al. [28] demonstrated that the diffracted peaks from the martensitic α′ phase (distorted hcp crystalline structure) can be formed in Ti–Zr samples, resulting from the Zr action in the decay of the Ms temperature for α′ phase precipitation. In our study, the binary Ti–Zr alloy was solubilized with a furnace cooling rate. However, it was fast enough to form this metastable phase, following the results reported by Correa et al. [28]. The diffraction peaks slightly shifted towards smaller angles compared to those of CP–Ti G4, which is attributed to the large atomic radius of Zr (160 pm) compared to Ti (147 pm). This dilates the hcp cell parameters and causes the peaks shifting according to the Bragg’s law. Correa et al. [29] affirm that the XRD profile of titanium in the hexagonal structure (α and α′ phase) is identical due to the same spatial group. According to Biswas et al. [30], the martensitic transition in titanium alloys is caused by deformation or phase transition in the cooling from high temperatures. As a result, in the case of Ti–Zr binary alloys, these conditions were well-established to obtain the acicular-shaped martensitic α′ phase in the matrix α phase. Therefore, this alloy can be indexed with the same α-Ti phase (JCPDS 00-044-1294 card).

The Ti–Zr–Mo shows a slight increase in the peak intensity at approximately 38°, indicating the presence of a secondary β-Ti phase. Besides this, the ternary material was submitted to a furnace cooling rate in the solution treatment, and it was fast enough to form metastable phases, α″- and β-phases. The β-phase corresponds to the body-centered cubic (bcc) β-Ti phase (JCPDF 01-089-3726 card). These peaks are expected to form by adding Mo, which lowers the phase transition temperature (β-transus) and the martensite start (Ms) temperature [31]. Correa et al. [28] demonstrated similar results regarding these metastable phases on this alloy, finding a biphasic composition of orthorhombic (martensitic α″ phase) and bcc (β-Ti phase) after the steps of homogenization heat treatment at 100 °C for 24 h with slow cooling, hot-rolling at 100 °C with air cooling, and quenching at 850 °C for 20 h with water cooling were all performed to the ingots.

### 3.4. Microstructural Characterizations

Based on the OM and SEM micrographs shown in Figure 3, it is possible to note that CP–Ti G4 was entirely composed of the α phase grains.

The microstructure of the Ti–Zr sample consisted of small acicular structures, typical of the martensite α′ phase. Some laths were permeated, which is related to the α phase. According to Biswas et al. [30], the acicular α′ and lath α phases constitute the microstructure of Ti–Zr binary alloys with randomly distributed and oriented structures.

According to Correa et al. [29], the Ti–15Zr–5Mo alloy presented irregular grains of β phase with the formation of fine needles of martensite α″ phase along with the intragranular region. In this study, Ti–Zr–Mo had a matrix of β phase in the microstructure, with some amount of martensite α″ phase probably formed during the cooling of the solubilization heat treatment.

### 3.5. Microhardness Tests

Figure 4a demonstrates the Vickers microhardness results obtained at points 1–13; Figure 4b shows the average microhardness values. The CP–Ti G4 depicts an average hardness of 201 ± 6, which is lower than that of the binary and ternary alloys. The Ti–Zr sample shows an average microhardness of 241 ± 9, whereas the Ti–Zr–Mo sample demonstrates the highest average hardness of 292 ± 8. The results indicate an increase in Vickers microhardness with addition of Zr and Mo, which are hardener alloying elements. Mo was revealed to be the hardest alloy material in recent research on Ti-based alloys for biomedical applications [32]. However, Correa et al. [29] suggested a nonlinear hardness behavior with the concentration of Mo and Zr, owing to the combination of the solid solution and phase precipitation hardening mechanisms.

Thus, two mechanisms could be proposed to explain the hardening of the samples: (1) the alloying elements (Zr and Mo) acted as hardeners in the solid-solution hardening, and (2) phase-precipitation hardening is caused by the metastable phases (α′, α″, and β).

### 3.6. Electrochemical Tests

The OCP, recorded until stabilization, is shown in Figure 5, and the average OCP values are listed in Table 2. The Ti–Zr–Mo sample exhibited a positive OCP value of 0.10 V. In contrast, the Ti–Zr and CP–Ti G4 alloys demonstrate negative OCP values of −0.01 V and −0.44 V, respectively. A positive OCP indicates a noble behavior and stable recovery of the natural TiO_2_ layer formed on the surface. This improvement in the natural equilibrium can be related to alloying elements Zr and Mo added to Ti, which supports stability.

When Zr is added to an alloy, the protective effect of Zr is a function of both the natural ZrO layer and the grain boundary where ZrO proliferates, thereby forming a protective layer between these boundaries and promoting a lower corrosion current density [33]. In addition, ZrO_2_ competes with TiO_2_, depending on how it is bound on the surface. The greater its quantity on the surface, the more excellent the corrosion resistance, owing to its high stability [34]. Xu et al. [35] reported that properties such as microhardness, corrosion potential, and corrosion current increased with the positive contribution of Zr concentrations up to 40 wt.%. Cardoso et al. [36] studied the effect of Mo addition on cell adhesion and viability and concluded that cell adhesion was low for the investigated alloys. However, the adhesion value of the binary Ti–5Mo alloy was observed to be the closest to that of Ti–6Al–4V than those of other alloys with 10% and 20% Nb. The binary alloy with Ti–5Mo and the alloys with Nb showed better viability for Ti–6Al–4V. Furthermore, Mo formed a thin, dense, and compact layer of passive oxide, slowing down the titanium dissolution process and favoring corrosion resistance [37].

The linear Tafel extrapolation method (Figure 5b) was used to evaluate the corrosion process of the samples. The ternary alloy shows the highest positive *Ecorr* values of 0.05 V during the corrosion process, followed by the binary alloy with a value of −0.12 V. The CP–Ti G4 exhibits the lowest *Ecorr* value of −0.60 V. The more positive the value of *Ecorr*, the better the electrochemical response compared to other samples. The corrosion current density (*Icorr*) was obtained from the intersection of the anodic and cathodic regions of the linear Tafel extrapolation, which is one of the primary variables to relate with corrosion. The lower the current density values, the better the corrosion resistance.

The ternary alloy with the lowest *Icorr* value of 6.13 × 10^−8^ mA·cm^−2^ is observed to be the most resistant to the corrosion process. The binary alloy and CP–Ti G4 samples exhibit *Icorr* values of 6.54 × 10^−8^ and 6.13 × 10^−8^ mA·cm^−2^, respectively. However, a high *RP* implies a high corrosion resistance. The *RP* acts as a resistor with an insulating effect owing to the natural TiO^2^ film formed by the reaction products on the sample’s surface. The binary and ternary alloys show lower *RP* values than the CP–Ti G4. However, all samples exhibit good values to be used in medicine.

High *CR* values are associated with a significant release of metallic ions and particles from the implant surface, which may affect the surrounding living cells and tissues, thereby, causing necrosis [38]. The release of ions and debris can increase in complex environments with friction during tribocorrosion. The Ti–Zr–Mo and CP–Ti G4 exhibit the same corrosion rate (*CR*) of 0.5 × 10^−3^ (mm·yr^−1^), whereas Ti–Zr shows a high *CR* of 0.6 × 10^−3^ (mm·yr^−1^). The values are low and significantly close in all the investigated alloys and other investigations in the literature; for instance, Gindre et al. [39] obtained a value of 0.3 × 10^−3^ (mm·yr^−1^) for Ti–CP, indicating the applicability of these alloys in medicine. Notably, the CP–Ti G4 is already commercially used in the case of dentistry for implants of dental abutments. Furthermore, it is utilized in the orthopedic case for the acetabulum head of hip implants.

Moreover, all the investigated samples depict the long passivation stages until 2 V, indicating that a protective layer forms on the surfaces of the samples owing to the well-known nearly linear stabilization of the passivation protective ability of Ti alloys. The CP–Ti G4 exhibits a stable zone up to 2 V, whereas a gradual increase in the current in the anodic region is observed in Ti–Zr and Ti–Zr–Mo alloys because of the film-dissolution. Therefore, materials with low *CR* should be used as implants to avoid the release of ions and debris into the bloodstream, which can cause allergies or toxicity in the body. Cui et al. [40] suggested that the addition of Zr improved the chemical-dissolution resistance of the TiO_2_ passivation film, and the addition of Mo formed a passivation film with excellent corrosion resistance on the alloy surface.

### 3.7. EIS Measurement

EIS was performed to evaluate the corrosion kinetics of the sample surfaces in contact with the electrolyte. The Nyquist diagram (Figure 6a) shows an open arch with an enlarged diameter in the CP–Ti G4, indicating more capacitive behavior than other investigated alloys. The Ti–Zr surface shows a small magnitude with adding Zr to the alloy. Furthermore, the Ti–Zr–Mo surface exhibits a more resistive behavior with a significantly higher magnitude, suggesting a superior impedance with higher resistance to ion exchange. Higher impedance values indicate enhanced electrochemical behavior [41]. The Bode diagram (Figure 6b) depicts similar decay curves and impedance values (|Z|) for all three investigated samples, indicating similar corrosive behavior in the electrolyte. This is excellent because the investigated alloys follow the same pattern as the commercial grade CP–Ti G4. In addition, the Ti–Zr–Mo surface shows a higher value of |Z| at all frequencies, with a tendency to increase at low frequencies compared to the other investigated alloys. The |Z| values at low and high angles indicate the surface and electrolyte impedances, respectively. Regarding the phase-angle plot, all the group curves follow a similar pattern, exhibiting a low phase angle at high frequencies with a tendency to increase between 10^3^ and 10^0^ Hz and decrease at low frequencies.

The data were modeled in an equivalent electrical circuit, as shown in Figure 7, for all samples. The calculated parameters are listed in Table 3. The equivalent circuit consists of an ohmic solution resistance (Rs) of the electrolyte, an experimental human body fluid solution (PBS) in series with a parallel arrangement of a constant phase element (CPE), and a resistor. The CPE represents a non-ideal capacitor owing to the inhomogeneous nature of the passive layer formed on the alloy surface. The resistor expresses the RP that, in this case, stands for the corrosion resistance of the natural passive layer, which acts as a damper to the flux of electrons, causing a resistance across the metallic material interface. PBS measures the level of protection provided by the natural passive layer which forms at the interface [42]. The closer the chi-squared value (χ^2^) is to 0, the better the fit. This study’s values are close to zero (~0.1), suggesting adequately fitted circuits. An RP value of 5.98 × 10^5^ Ω is observed for Ti–Zr, followed by that of CP–Ti G4 (5.04 × 10^5^ Ω).

The charge transfer coefficient (α) was calculated to deduce the kinetics of the electrochemical reaction. The α, close to 1, describes the capacitive behavior of the samples; A value of α close to 0 indicates a resistive behavior. The CP–Ti G4 shows higher capacitance behavior with α = 0.93, which is consistent with the open arch in the Nyquist diagram. Ti–Zr and Ti–Zr–Mo exhibit lower capacitive behavior than that of CP–Ti G4 because of the natural protective layer formed on the surface. Therefore, a significant improvement is observed by adding Zr and Mo in the electrochemical properties obtained from the potentiodynamic curves of both binary and ternary alloy surfaces. The test results follow the sequence of the samples, starting from the best behavior in polarization resistance of the Ti–Zr–Mo, followed by Ti–Zr, to the lowest performance of CP–Ti G4, which is used in biomedical applications.

### 3.8. Tribocorrosion Tests

The tribocorrosion analysis of each alloy was performed to evaluate the variation in the OCP. The COF for different alloys is shown in Figure 8, and the values are listed in Table 4. In addition, results are plotted in bar graphs to compare the results obtained for the investigated alloys. Figure 8a demonstrates that the initial potential of CP–Ti G4 in the stabilization zone is −0.36 V before the sliding begins, which is lower than that of the other investigated alloys, indicating a higher reactivity. This result agrees with the OCP results presented in Section 3.5.

During the sliding, this potential drops to −0.81 ± 0.04 V, where the standard deviation represents slight oscillations, indicating a fast recovery with the tribocorrosion scratches. The observations agree well with the literature [43]. The potential fluctuations are related to the phenomena of destruction and recovery of the passive layer during the process. Once the sliding ends in the passivation zone, the potential rapidly recovers to a value of −0.30 V, similar to the OCP in the stabilization zone. This indicates the stability of Ti with the recovery of total oxide after wear damage owing to the formation of a new barrier on the surface against corrosion [44]. Furthermore, OCPs of the binary and ternary alloys are observed to return to higher values than those of CP–Ti G4 after sliding, indicating a better passivation capacity of the film.

The Ti–Zr exhibits better potential, stabilized at −0.04 V, than the CP–Ti G4. Furthermore, the OCP of the binary alloy instantly decreases when the sliding starts, thereby exhibiting a higher standard deviation (0.45 + 0.35 V) of the total OCP variation (|Δ| OCP) than all the investigated alloys. However, the binary alloy reaches only −0.13 V in the passivation zone, which is 0.14 V lower than the initial potential achieved in the stability zone. In addition, Barão et al. [45] suggested that the Ti–5Zr alloy exhibited better tribocorrosion and mechanical strength than CP–Ti G4. In contrast, the Ti–Zr–Mo depicted only the positive OCP potential with 0.04 V, higher than the other investigated alloys, indicating a lower reactivity. Buciumeanu et al. [2] reported an abrupt cathodic change in the potential and mechanical action of the alumina sphere during the sliding process, which removes the TiO_2_ film, naturally owing to the significant changes in the electrochemical properties. However, the ternary alloy reaches only −0.27 V in the passivation zone, which is 0.31 V lower than the initial potential in the stability zone. Thus, due to the stable OCP, the Ti alloy cannot recover its passive layer in the passivation zone. Nevertheless, the Ti–Zr–Mo potential recovers the initial stability faster (−0.27 V) than the Ti–Zr, which reaches a value of −0.13 V in the stabilization zone within 3000 s. Thus, the Ti–Zr–Mo and Ti–Zr alloys exhibit better wear potentials than CP–Ti G4.

Despite the higher OCP values to Ti–Zr and Ti–Zr–Mo, both alloys show a higher potential variation of 0.90 ± 0.07 V and 0.72 ± 0.06 V, respectively, during the sliding process, without perceptive variation on the standard deviation. A higher variation in the sliding potential indicates the subsequent recovery and destruction of TiO_2_ during all sliding zones. Pohrelyuk et al. [46] suggested that the original oxide film was removed from the metal surface during the sliding process when the high point of the layer hardened the oxide, leaving the underlying metal uncovered. The new metal reacted quickly with oxygen from the air to form a new oxide layer, which was removed in the next cycle. These sequences of events are observed in the passivation of the binary and ternary alloys in this study.

Furthermore, the mechanical COF was investigated for the alloys based on the tribocorrosion tests, which indicate the sliding resistance between sample surfaces and their alumina counterparts. The COF obtained during sliding is shown in Figure 8c. The CP–Ti G4 exhibits the lowest resistance, with an average COF of 0.38 ± 0.07. The Ti–Zr is in second place, with a COF of 0.39 ± 0.03, followed by the Ti–Zr–Mo, with an average COF of 0.44 ± 0.05. The COF value, in general, follows the same trend as the hardness does. The harder the alloy, the higher the COF. Therefore, the trend observed for the COF is consistent with the microhardness data, shown in Section 3.6.

With the addition of Mo, the resistance to tribocorrosion improves between 8 and 16 wt.% using powder metallurgy. These values are close to those of Ti–6Al–4V. In addition, the investigated alloy does not have any toxic metals. However, Xu et al. [35] reported that values above 20 wt.% of Mo decreased the resistance to tribocorrosion; however, they exhibited values that were better than those of CP–Ti. In addition, the higher COF value of the ternary alloy could be associated with the constant passivation tendency during sliding [43,47], which could hinder debris release during the wear process [48,49].

The topographies of the wear track of the titanium alloys were compared after tribocorrosion (Figure 9). The lines on the channels indicate an abrasive wear mechanism, characterized by the material detachment provenance from wear damage [50]. The topographical and chemical analyses of Ti–Zr and CP–Ti G4 samples show several chemical and morphological similarities [51]. The confocal images demonstrate a slightly more worn track in the Ti–Zr alloy than in the CP–Ti G4, which agrees with a higher coefficient of friction for Ti–Zr alloy. According to Alves et al. [52], once the anodic oxide film is worn out, the wear becomes higher than the wear measured in pure titanium under the same testing conditions. This is because of third-body mechanisms resulting from abrasive oxide wear debris, which remain in the contact area [53]. The confocal topography indicates the presence of this third-body mechanism on the edge of the track, circled in the Ti–Zr alloy (Figure 9). In contrast, the Ti–Zr–Mo alloy shows the highest COF among the investigated alloys. In addition, the track exhibits a lower thickness, indicating less wear than the other investigated alloys owing to the elevated hardness of the ternary alloy. Toptan et al. [54] claimed that samples with higher OCP values in the three sliding zones (before, during, and after sliding) exhibited enhanced corrosion resistance due to oxide layers produced on the surfaces and their high hardness, which also increased wear resistance. The order in which the samples perform, owing to the thinner layer seen in the confocal microscopy images, is Ti–Zr–Mo > Ti–Zr > CP-Ti G4.

Figure 10a shows the two-dimensional (2D) topography profile of the wear tracks after tribocorrosion was used to calculate the volume loss and, consequently, the wear rate. Figure 10b demonstrates the volume loss of the investigated alloys, indicating that (1) the ternary alloy depicts a significantly higher volume loss than the other alloys; and (2) the average depth of the wear track is coherent with the COF, following the same order of CP–Ti G4 < Ti–Zr < Ti–Zr–Mo. In addition, Figure 10c shows the obtained data from the total energy during the 1800 s of the tribocorrosion tests. In the case of the Ti–Zr–Mo alloy, despite the higher volume loss (96.3 × 10^3^ mm^3^), it was necessary to use more energy (5.51 N·m) to remove material during tribocorrosion, mainly due to hardness properties. Therefore, the wear rate of Ti–Zr–Mo showed a slightly better performance than that of Ti–Zr, as shown in Figure 10d.

Therefore, the hardness, OCP, COF, wear track, and volume loss values of both developed samples were studied and compared in tribocorrosion with the well-known excellent properties of pure titanium. The values are approximated and sometimes overlap with CP–Ti G4 behavior.

## 4. Conclusions

To wrap up, the following conclusions were drawn for the investigated alloys:

Corrosion tests revealed that adding Zr to Ti enhanced corrosion behavior by boosting potential corrosion values. Adding Mo to this binary alloy improved corrosion behavior even further. Moreover, the Ti–Zr–Mo alloy showed the lowest corrosion current density value. As a result, it demonstrated better corrosion behavior than the other alloys under investigation. Furthermore, the Ti–Zr–Mo sample had a significantly greater magnitude arch and less capacitive behavior, indicating superior electrochemical performance.

Despite having the highest coefficient of friction and volume loss of the investigated samples, the Ti–Zr–Mo alloy required more energy to remove the material during tribocorrosion, due to its high hardness. As a result, Ti–Zr–Mo had a marginally more significant wear rate than Ti–Zr. Furthermore, the ternary alloy had a narrower track visible in the confocal microscope images, indicating that it has the intense recovery capability of the natural oxide layer throughout the experiments based on the open circuit potential, which contributes to the outstanding tribocorrosion behavior.

Due to those factors, it was assumed that these newly developed biomaterials would exhibit superb behaviors by closely resembling the excellent qualities of pure titanium or occasionally overriding the behavior of CP–Ti G4. Hence, both the binary and ternary alloys are great candidates as biomaterials to overcome the drawbacks associated with corrosion and tribocorrosion during long-term usage in the human body.

## Figures and Tables

**Figure 1 materials-16-01826-f001:**
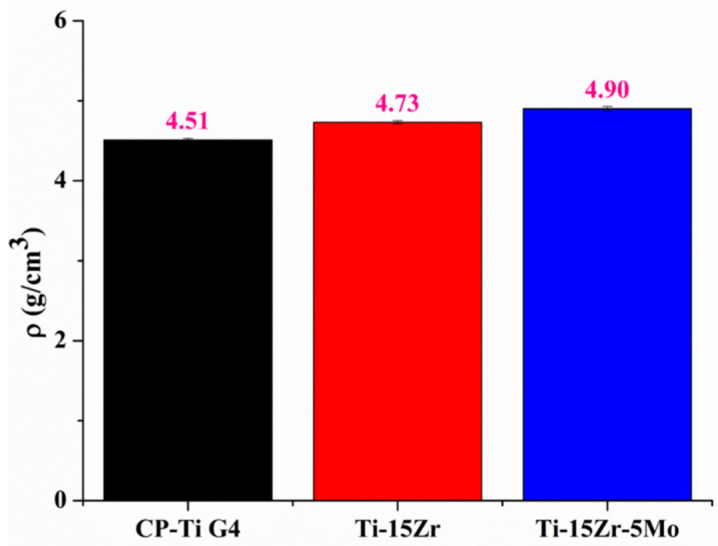
Comparison of density values.

**Figure 2 materials-16-01826-f002:**
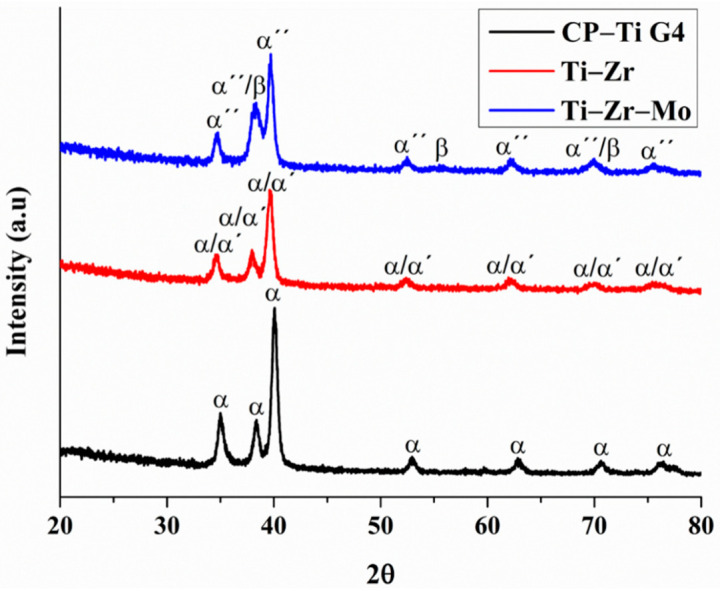
X-ray diffraction patterns of the samples with α (alpha phase), α′ (martensitic alpha phase), α″ (metastable alpha phase) and β (beta phase) identified.

**Figure 3 materials-16-01826-f003:**
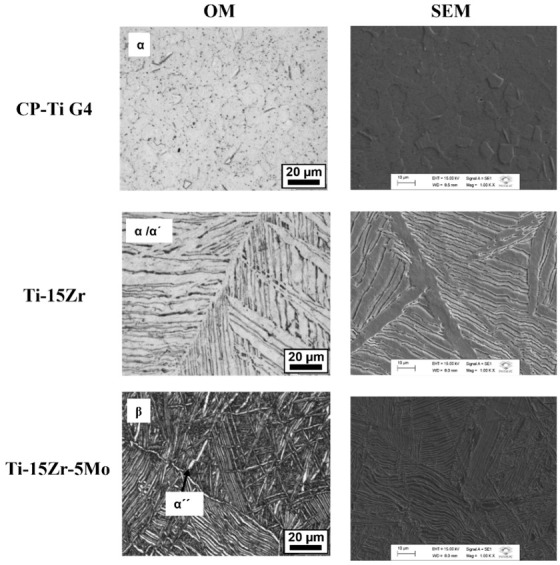
OM and SEM of the samples.

**Figure 4 materials-16-01826-f004:**
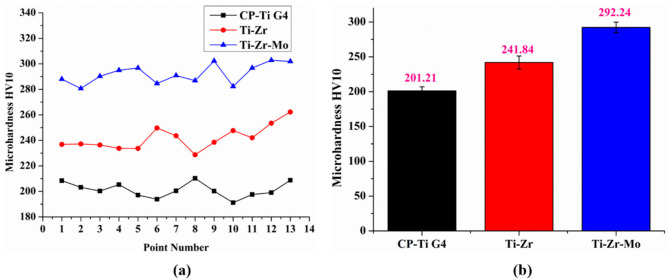
(**a**) Vickers microhardness by point numbers. (**b**) bar plots of average values of Vickers microhardness.

**Figure 5 materials-16-01826-f005:**
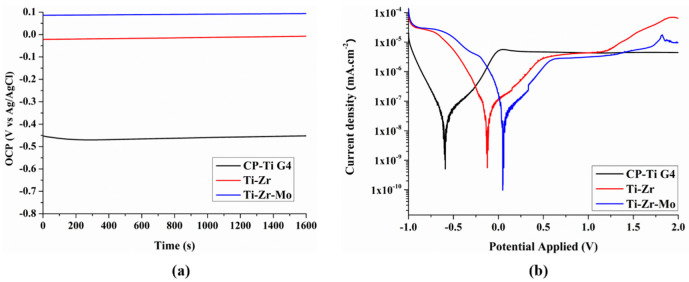
(**a**) open-circuit potential (OCP) (**b**) linear Tafel extrapolation in PBS.

**Figure 6 materials-16-01826-f006:**
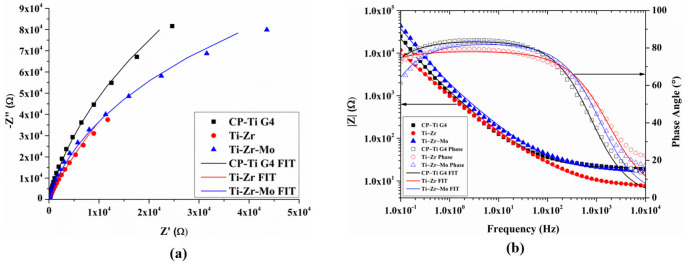
(**a**) the Nyquist diagram and (**b**) the Bode diagram.

**Figure 7 materials-16-01826-f007:**
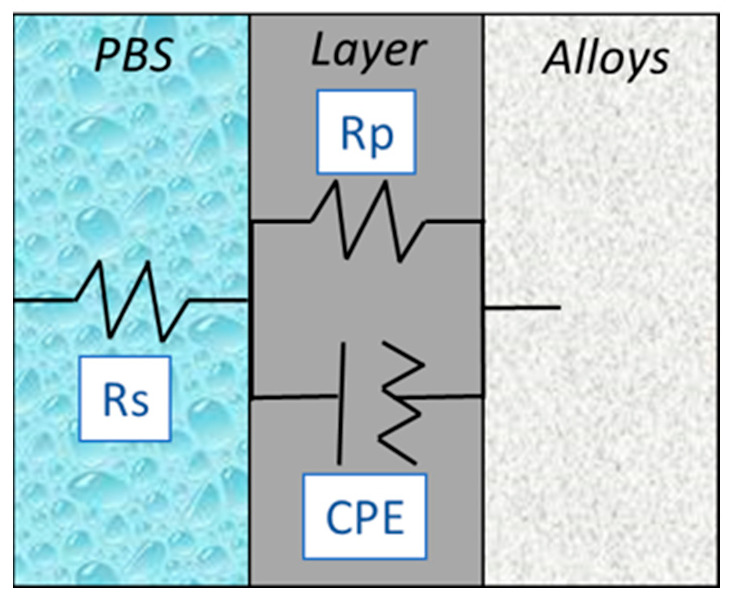
The equivalent electrical circuit model.

**Figure 8 materials-16-01826-f008:**
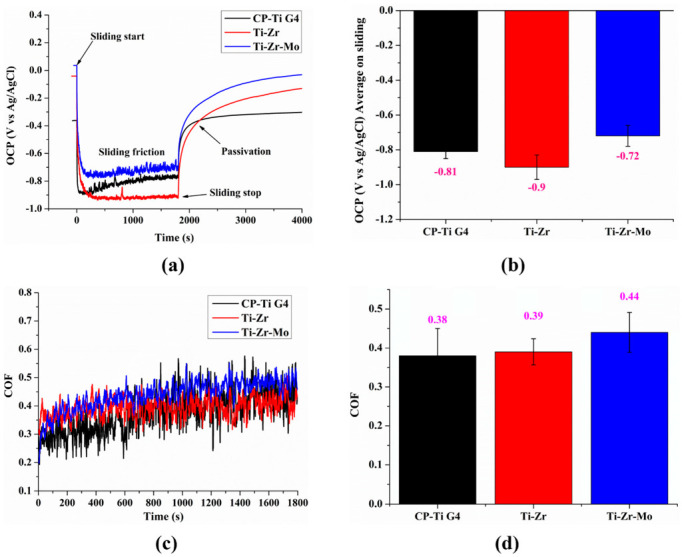
(**a**) open-circuit potential (OCP vs. Ag/AgCl) before, during, and after tribocorrosion tests, of all the investigated alloys. (**b**) bar plots of average values on sliding. (**c**) coefficient of friction (COF) during sliding. (**d**) bar plots of COF average.

**Figure 9 materials-16-01826-f009:**
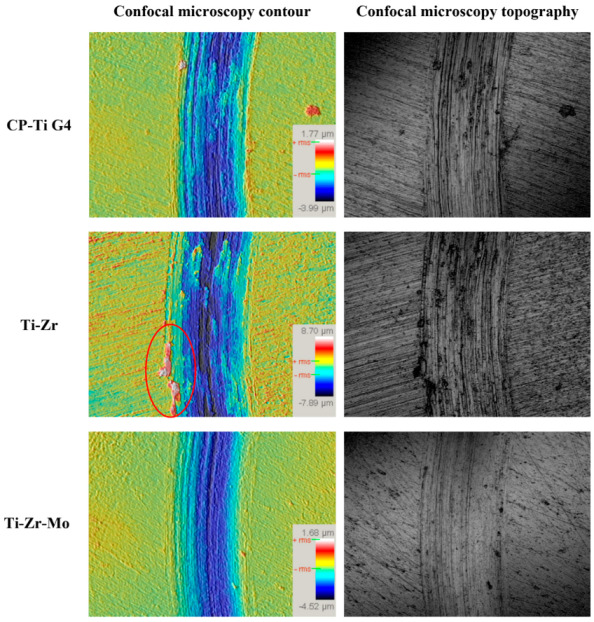
Profiles obtained by confocal microscopy 100× in the central region of the wear tracks.

**Figure 10 materials-16-01826-f010:**
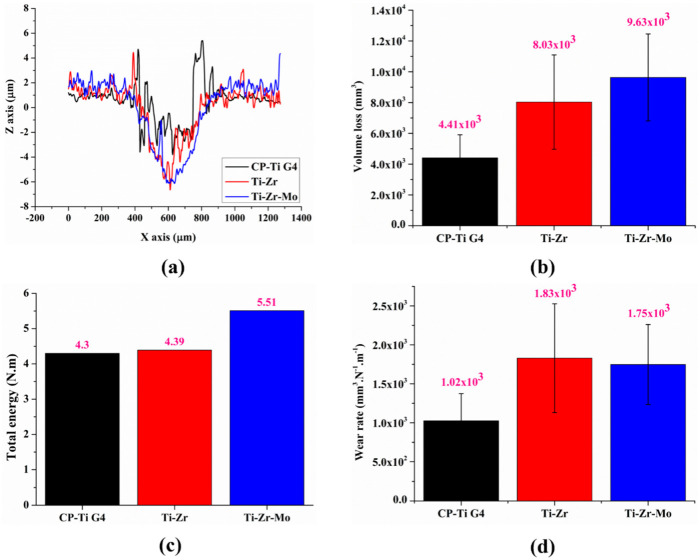
(**a**) profiles in X and Z axes. (**b**) bar plots of volume loss. (**c**) bar plots of total energy. (**d**) bar plots of wear rate.

**Table 1 materials-16-01826-t001:** Chemical analysis of samples content (wt.%).

Samples	Ti	Fe	Zr	Mo
CP–Ti G4	99.78	0.22	-	-
Ti–Zr	84.98	-	15.02	-
Ti–Zr–Mo	77.80	-	15.90	6.30

**Table 2 materials-16-01826-t002:** Electrochemical parameters obtained from linear polarization curves of the alloys studied in PBS.

Alloys	OCP (V)	*E_corr_* (V)	*I_corr_* (mA·cm^−2^)	*R_P_* (Ω·cm^2^)	*CR* (mm·yr^−1^)
CP–Ti G4	−0.44	−0.60	6.82 × 10^−8^	1.05 × 10^6^	0.5 × 10^−3^
Ti–Zr	−0.01	−0.12	6.54 × 10^−8^	7.77 × 10^5^	0.6 × 10^−3^
Ti–Zr–Mo	0.10	0.05	6.13 × 10^−8^	5.26 × 10^5^	0.5 × 10^−3^

**Table 3 materials-16-01826-t003:** Parameters obtained from the equivalent circuit model.

Element	CP–Ti G4	Ti–Zr	Ti–Zr–Mo
R_s_ (Ω)	20.06	7.73	17.67
R_p_ (Ω)	5.04 × 10^5^	5.98 × 10^5^	2.73 × 10^5^
CPE (C)	1.81 × 10^−5^	3.92 × 10^−5^	1.60 × 10^−5^
α	0.93	0.87	0.92
χ^2^	0.15	0.16	0.18

**Table 4 materials-16-01826-t004:** Electrochemical and tribological parameters, obtained from the potentiodynamic polarization curves of the alloys studied in PBS.

Alloys	OCP (V) before Sliding	OCP (V) Average on Sliding	|Δ| OCP Total (V)	Passivation (V)	COF
CP–Ti G4	−0.36	0.81 ± 0.04	0.53 ± 0.24	−0.30	0.38 ± 0.07
Ti–Zr	−0.04	0.90 ± 0.07	0.45 ± 0.35	−0.13	0.39 ± 0.03
Ti–Zr	−0.04	0.90 ± 0.07	0.45 ± 0.35	−0.13	0.39 ± 0.03

## Data Availability

Data will be available upon reasonable request.

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
