# Peer review of "Ti–15Zr and Ti–15Zr–5Mo Biomaterials Alloys: An Analysis of Corrosion and Tribocorrosion Behavior in Phosphate-Buffered Saline Solution"

_materials, 2023, doi:10.3390/ma16051826_

Round 1

Reviewer 1 Report

The manuscript is devoted to the study of the corrosion and tribocorrosion behavior of three titanium alloys: two groups of samples developed via arc melting (Ti–15Zr and Ti–15Zr–5Mo) and a control group of commercial alloy (CP–Ti G4). The Ti–15Zr and Ti–15Zr–5Mo  alloys are offered for use in the biomedical field. The addition of Mo to the Ti–Zr alloy increased the microhardness. The ternary alloy exhibited the better corrosion behavior than the other investigated alloys. In addition, the Ti–Zr–Mo showed the lowest Icorr value and the highest electrochemical response. The addition of Zr to Ti improved the chemical-dissolution resistance of the TiO2 passivation film.

            The manuscript is of great theoretical and practical interest, however there are some questions.

i. Doesn't the higher volume loss and higher wear rate of Ti–Zr and Ti–Zr–Mo alloys during tribocorrosion tests compared to CP–Ti G4 indicate their inferior properties?(Figure 8). Doesn't this mean that when chewed, these new biomaterials used for dental implants will wear out faster than CP-Ti G4?

ii. Page 12, 4-5 lines from the bottom: "[40] indicated that the samples in which showed higher OCP values before, during, and after sliding." Authors should explain this phrase

Reviewer 2 Report

1) Grammatically title of the article is wrong. Please revise it.

2) Introduction must be revised, it has serious amount of similarity index.

3) Authors must cite the recently published articles in the introduction either in the table or in the text. It is recommended to cite the below articles related to titanium alloys, 

https://doi.org/10.1016/j.powtec.2022.117715, https://doi.org/10.33263/BRIAC132.118

4) Materials and methods section must be re-written to reduce the similarity index.

5) Figure 1, XRD is confusing. Authors has got the peaks at same bragg's angle in all the 3 samples. But for Ti-Zr-Mo samples author mentioned β-phase, but at the same diffraction angle authors mentioned α-phase for Ti-Zr and CP–Ti G4. Why?

6) Authors mentioned that presence of β-phase in Ti-Zr-Mo sample could be the one reason for increased hardness, but i recommend to perform optical microscope or SEM to show both α and β-phase. XRD alone is not enough to show the phases.

7) Tables are not according to the journal requirement.

8) Authors must discuss the wear mechanism in detail to understand the wear properties in detail. It is better to mention the wear modes and the wear mechanism in the figure 7.

9) The more discussion on biomaterial application is missing in the results and discussion.

Reviewer 3 Report

The present study investigated Ti alloys' corrosion and tribocorrosion behavior in a simulated biological environment. The paper is obviously of interest to researchers working in this field. However, the manuscript should be amended before its acceptance. I want to address the following issues:

1. Abstract. There is an error in the sentence, “The primary objective of this study was to investigate the corrosion and tribocorrosion behavior of three titanium alloys: two groups of samples developed via arc melting (Ti–15Zr and Ti–15Zr–5Mo) and a control group of commercial alloy (CP–Ti G4)”. Ti Grade 4 (CP-Ti G4) is not an alloy but is pure titanium. The titanium alloys were, therefore, two. A similar error occurs in the Introduction (last paragraph).

2. The paper lacks microscopic analysis of the produced alloys (Ti-15Zr, Ti-15Zr-5Mo) and pure titanium. Please add photos of the microstructure of the tested materials. It was also good to include the final chemical composition of the materials.

3. Chapter 2.5. Vickers measurements at a load of 10 kgf are not microhardness but hardness. This is too high a load to talk about microhardness.

4. Figure 1 is too small and of poor quality.

5. In Fig. 2 (a and b), there is an error in describing the Y axis. Please remove the parenthesis, as HV is not the unit of hardness measurement by the Vickers method, but only its designation. In the description of Fig. 2, there is also the same error described in point 3 of this review.

6. The results of tribological tests should also be presented using a volumetric wear index [mm3/Nm].

Round 2

Reviewer 2 Report

Authors have improved the manuscript as per my comments. Thank you